# The OvarianTag™ Biomarker Panel Emerges as a Prognostic Tool to Guide Clinical Decisions in Cisplatin-Based Treatment of Epithelial Ovarian Cancer

**DOI:** 10.3390/ijms26178393

**Published:** 2025-08-29

**Authors:** Letícia da Conceição Braga, Laurence Rodrigues do Amaral, Pedro Henrique Villar Delfino, Nara Rosana Andrade, Paulo Guilherme de Oliveira Salles, Agnaldo Lopes da Silva Filho, Pedro Luiz Lima Bertarini, Ana Paula Álvares da Silva Ramos, Matheus de Souza Gomes, Luciana Maria Silva Lopes

**Affiliations:** 1Oncotag Desenvolvimento de Produtos e Servicos Para Saúde HumanaRua Tiradentes, 2689, Sala 102, Industrial, Contagem 32230-020, MG, Brazil; pedro.villar@oncotag.com.br (P.H.V.D.); nara.oncotag@gmail.com (N.R.A.); anapalvares@gmail.com (A.P.Á.d.S.R.); 2Laboratório de Pesquisa Translacional em Oncologia, Núcleo de Ensino, Pesquisa e Inovação, Instituto Mário Penna, Belo Horizonte 30380-420, MG, Brazil; sallespgo@gmail.com; 3Universidade Federal de Uberlândia, Campus Patos de Minas, Patos de Minas 38702-178, MG, Brazil; bertarini@ufu.br (P.L.L.B.); souzagomes.matheus@gmail.com (M.d.S.G.); 4Departamento de Ginecologia e Obstetrícia, Faculdade de Medicina, Universidade Federal de Minas Gerais, Belo Horizonte 31270-901, MG, Brazil; agnaldo.ufmg@gmail.com; 5Serviço de Biologia Celular, Diretoria de Pesquisa e Desenvolvimento, Fundação Ezequiel Dias, Belo Horizonte 30510-010, MG, Brazil; luciana.silva@funed.mg.gov.com

**Keywords:** epithelial ovarian cancer, extrinsic apoptosis cell signaling, platinum-resistance, OvarianTag™, biomarkers

## Abstract

Epithelial ovarian cancer (EOC) is the most lethal gynecologic malignancy, often diagnosed at an advanced stage due to its asymptomatic progression. The high recurrence rate and development of platinum-based chemotherapy resistance contribute to its poor prognosis. Despite advancements in molecular profiling, predictive biomarkers for chemotherapy response and recurrence risk remain limited. In this study, we developed OvarianTag™, a biomarker panel integrating apoptosis and necroptosis pathways, to predict chemotherapy benefit and disease progression in EOC patients. This observational study was conducted in two phases. In the first phase, 45 patients were recruited, and RNA was extracted from fresh ovarian tissues (normal, benign, and malignant). qRT-PCR was performed to assess the relative expression of genes involved in apoptosis and necroptosis-regulated cell death pathways. Machine learning algorithms were applied to identify the relevant prognostic markers, leading to the development of OvarianTag™. In the second phase, 55 additional EOC patients were included, and their formalin-fixed, paraffin-embedded (FFPE) tumor samples were analyzed using qRT-PCR. The classifier algorithm incorporated hierarchical clustering to stratify patients based on gene expression profiles. Significant differences in *TNFRSF10C/TRAIL-R3*, *TNFRSF10B/TRAIL-R2*, and *CASP8* expression levels were observed between patient groups. *CASP8* downregulation was strongly correlated with platinum resistance and a poor prognosis. Decision tree models achieved 83.3% accuracy in predicting platinum response and 79.2% accuracy in recurrence risk stratification. The OvarianTag™ classifier demonstrated high sensitivity and specificity in identifying high-risk patients, supporting its potential as a prognostic tool. The OvarianTag™ panel provides a novel approach for risk stratification in EOC, integrating apoptosis and necroptosis pathways to refine chemotherapy response prediction and recurrence risk assessment. This molecular assay has the potential to guide personalized treatment strategies, enhancing clinical decision-making and improving patient outcomes. Further validation in independent cohorts is warranted to establish its clinical utility.

## 1. Introduction

Ovarian cancer (OC) ranks seventh among malignant tumors and eighth as a cause of cancer-related death in women worldwide, presenting an average five-year survival rate of 43% of cases. OC accounted for nearly 313,959 cases and 207,252 deaths in 2020 (7.1% incidence and 5.8% mortality) [1]. Furthermore, current estimates reveal alarming data: 19,710 new cases, of which 13,270 are expected to result in death [2]. In Brazil, according to data from the National Cancer Institute (INCA), 7310 new cases of OC are estimated for the years 2023–2025 [3]. Although OC is considered a low-incidence cancer compared to others, such as breast and cervical cancers, it has the highest mortality rate among malignant diseases affecting women [4,5].

Despite these alarming statistics, the diagnosis of ovarian cancer remains particularly challenging due to its insidious nature, posing a significant challenge in gynecologic oncology [6]. Another critical factor is that OC can be divided into subtypes according to the origin of the tumor cells, and these main histological subtypes can be characterized as serous carcinoma (high and low grade), endometrioid, mucinous, and clear cell carcinoma [7]. Among the ovarian cancer subtypes, high-grade serous carcinoma (HGSC) is the most aggressive and lethal, accounting for up to 80% of cases. Frequently diagnosed at an advanced stage, this subtype is associated with a poorer prognosis, a high rate of cellular proliferation, a greater propensity for resistance to chemotherapy, and a higher recurrence rate [8].

Given the aggressive nature of HGSC and its association with poor prognosis, early detection is crucial, as it significantly improves survival rates [9]. In addition to diagnosis, defining the best therapeutic strategies available is also essential. Significant progress in radical debulking surgery and the advent of several successive lines of chemotherapy, including new cytotoxic drugs and maintenance therapy with targeted agents, have changed the cure rates of epithelial ovarian cancer (EOC) over the last five decades [10]. However, the origin and pathogenesis of the disease remain poorly understood [10,11].

On the other hand, chemotherapy remains a key element in OC treatment, often administered as a combination of a platinum compound, such as cisplatin or carboplatin, and a taxane, such as paclitaxel [12,13]. Although approximately 80% of patients show an initial positive response to standard chemotherapy, around 70% experience disease recurrence, with this percentage being even higher among patients in stages III–IV [14].

Thus, the effects of systemic chemotherapy for OC remain far from optimal, as most patients relapse within 18 to 24 months and develop resistance to various chemotherapy options, contributing to the poor prognosis of the disease [15].

Given these data, it is essential to explore and understand new potential prognostic tools based on biomarkers in OC. CA125 is the primary biomarker used in the routine diagnosis and monitoring of OC patients [16,17]. Predicting disease-free survival and overall survival remains a challenge for gynecologists and oncologists. Furthermore, one of the most pressing questions in current EOC therapy is identifying which patients will benefit from platinum-based chemotherapy. Developing a predictive test for such outcomes remains a critical unmet need. To provide more objective information and support more personalized clinical decisions, molecular tests have emerged as powerful tools to address this issue [18]. To date, no available test has proven capable of reliably predicting platinum-based chemotherapy response.

Special attention has been given to genetically determined cell death mechanisms, especially apoptosis and necroptosis, to better understand and overcome the challenge of chemoresistance in epithelial ovarian carcinoma (EOC) [19]. Apoptosis is a well-characterized programmed cell death pathway crucial to eliminating damaged or abnormal cells during development to maintain homeostasis [20]. In contrast, necroptosis is a programmed, regulated, and cysteine protease-independent cell necrosis that morphologically exhibits the same characteristics as necrosis. It occurs through a unique programmed cell death mechanism that is distinct from the apoptotic signaling pathway or when the apoptotic signaling pathway is inhibited [21]. Recent studies have demonstrated that dysregulation of these pathways significantly contributes to tumor progression and resistance to therapy, including platinum-based chemotherapy, in several cancer types, such as EOC [22].

In the present study, we describe a set of genes related to apoptosis and necroptosis and a machine learning algorithm capable of predicting tumor recurrence and the platinum response in EOC patients following surgery and subsequent chemotherapy treatment. To achieve this, we used an algorithm trained with EOC patient data and developed a molecular test named OvarianTag™.

## 2. Results

### 2.1. Individuals

Table 1 shows the clinical and pathological characteristics of the cohort. The only statistically significant difference between the groups was in the status of menopause. In the EOC group, the stage (FIGO) was I/II in five patients (31.8%) and III/IV in eleven patients (68.8%). The tumor was moderately differentiated (G2) in six cases (37.5%) and poorly differentiated in ten cases (62.5%).

In addition, another 55 patients were included. None of the clinical characteristics presented statistical differences except menopause status (*p* < 0.05).

Appendix A shows the distribution frequency of clinical and pathological EOC sample characteristics, including the rates of overall survival (OS) and disease-free progression (DFP). The mean ca-125 level was 1927.4 U/mL +3856.4, DFP 15.34 +6.8, and OS 23.7 +37 of the patients included in the second phase of this study.

### 2.2. Data of Gene Expression and Clinical Features Related

Our group has previously published expression profiles of the apoptosis target genes, *TNFRSF10C/TRAIL-R3, TNFRSF10B/TRAIL-R2,* and *BCL2*, as found across the patient groups [23,24,25].

Regarding *TRAF2*, *RIPK1*, *TNFRSF10D/TRAIL-R4*, and *NF-ΚB*, differential mRNA expression profiles were identified between the cystadenoma and EOC groups, with statistically significant differences observed between primary and metastatic EOC (Figure 1). A heatmap was constructed to provide a graphical representation of the gene expression data across the clinical and pathological characteristics of patients to analyze and visualize these expression patterns further (Figure 2).

### 2.3. Predictive Algorithm Designer to Clinical Outcomes Prediction in EOC Patients and Biological Relevance of Attributes Selected

The selection of biomarkers was performed using the feature selection inherently performed by the J48 method, an implementation of the C4.5 algorithm available in the WEKA software suite (3.6.11 version) [26,27]. The J48 algorithm was executed with the default input parameters and applied to datasets containing numerous attributes, including histological type, CA-125 levels, cancer stage, presence of ascites, tumor differentiation grade, cytoreduction status, progression-free survival, overall survival, and the expression of genes such as *TNFRSF10B, TNFRSF10C, TNFRSF10D, CASPASE8, BID, BAX, BCL2, APAF, RIPK1, TRAF2, TRAD1, NFKB*, and *TANK*. The J48 algorithm selected the most informative features based on information gain, excluding less relevant variables during the recursive partitioning process. After evaluating classification performance, only three features were retained based on their superior predictive value. All of the tested features, including those excluded and their respective performance metrics, are detailed in Appendix A. The chosen features were able to correctly classify the samples with at least 80.0% accuracy in full training and at least 90.0% accuracy in classifying new, previously unseen patients in LOOCV (Table 2). Notably, the J48 algorithm identifies the optimal cut-off points for separating the analyzed groups according to prognosis features (Figure 3).

With these results, three patent applications (BR1020160059151, BR1020160045541, and BR 10 2015 032299-2) were deposited in the Brazilian National Institute of Intellectual Property to support the technology, which is named the OvarianTag™ panel.

In the second phase, the second cohorts were enrolled. The gene expression profile obtained for three genetic biomarkers of the OvarianTag™ panel and clinical- pathological characteristics obtained from medical records were used for the analytical validation and training of the second model. In this phase, the J48 algorithm provided the prognostic value of the OvarianTag™ panel, defining which genes were used as attributes in the data mining and the threshold in gene expression to predict clinical outcomes (Figure 4). The method reports an accuracy of 86.7 and 83.3%, respectively, in determining the benefit of platinum-based treatment and the probability of tumor recurrence in less than 12 months (Table 3).

In the cross-validation, samples were also correctly classified to determine the benefit of the platinum-based treatment and the risk of tumor recurrence in 12 months, with an accuracy of 83.33% and 79.2%, respectively. It is worth mentioning that the accuracy parameters for predicting early and advanced EOC tumors are shown as the first model in both full training and LOOCV. All performance parameters presented were derived from cross-validation, as shown in Table 3.

The discriminative performance of the binary classifier was evaluated using the ROC curve and the corresponding Area Under the Curve (AUC) metric. We applied Leave-One-Out Cross-Validation (LOOCV); the model was trained on all but one observation and the excluded instance was used for testing. For each iteration, the predicted probability for the positive class was recorded alongside the true label. At the end of the process, all probabilities and ground truth values were aggregated to generate the overall ROC curve and compute the AUC. This procedure provides a robust estimation of the model’s ability to distinguish between classes, especially valuable in small-sample scenarios. The Figure 5 shows the ROC curve and the AUC value.

Statistically significant differences were observed for targets of the *OvarianTag*™ panel regarding primary and metastatic EOC groups (*p* = 0.025), ca-125 qualis (*p* = 0.031), menopause (*p* = 0.017), and DFS (*p* = 0.03), justifying the biological relevance of these biomarkers (Figure 6).

## 3. Discussion

In this study, we described an innovative tool, named OvarianTag™, designed to permit a better understanding of a prognosis in EOC and provide an opportunity to guide clinical intervention that could increase EOC survival rates. EOC is a disease characterized by high rates of relapse after standard chemotherapy and therefore requires novel strategies to overcome chemotherapy resistance [28]. The most active therapeutic agents are platinum analogs, such as cisplatin or carboplatin. Recurrence after initial platinum-based chemotherapy is standard among women diagnosed with advanced-stage cancer, with platinum resistance posing a significant challenge [29,30]. Several mechanisms contribute to the development of drug resistance, including tumor heterogeneity, reduced drug concentration at the target, alteration in drug target structure, and increased repair of the lesions induced [31,32,33]. A better understanding of the different resistance mechanisms is needed to improve the outcome for cancer patients [34]. Considering the potentially relevant factors that could influence gene expression and clinical outcomes, we evaluated clinical features consistently documented across all groups in medical records. Only menopausal status showed a statistically significant difference between groups. This evidence permitted us to affirm that the OvarianTag™ addresses these challenges by identifying gene expression patterns associated with platinum resistance, highlighting TNFRSF*10B*, *TNFRSF10C*, and *CASP8* as key attributes in its predictive algorithm. These genes are crucial in regulating cell death, as they are integral to apoptosis and necroptosis pathways.

The model achieved an AUC (Area Under the ROC Curve) of 0.8214 using Leave-One-Out Cross-Validation. This value indicates a good level of discriminative ability, suggesting that the model is capable of distinguishing between the two classes with high accuracy. An AUC above 0.8 is generally considered strong in binary classification problems, especially in domains where data availability is limited. The result also reflects a favorable balance between sensitivity and specificity across various classification thresholds. Given that LOOCV provides an almost unbiased estimate of generalization performance, the obtained AUC reinforces the reliability and robustness of the proposed model.

Apoptosis, also known as programmed cell death, is fundamental for the development and homeostatic maintenance of complex biological systems. The apoptotic process can be executed intracellularly by the release of several factors from the mitochondria (intrinsic pathways) or through the transmembrane death receptors (DRs), which are activated upon binding of their cognate ligands (extrinsic pathways). One is TRAIL, a TNF-related apoptosis-inducing ligand that is a type II transmembrane protein. Apoptosis can also be induced by a T-cell-mediated cytotoxicity pathway dependent on the perforin granzyme [35]. The human TRAIL-Rs can be subdivided into two classes: the full-length intracellular Death Domain (DD-containing receptors) TNFSF10A and TNFSF10B, which can induce apoptosis and are most widely expressed, and the alternative decoy receptors TNFSF10B and TNFSF10D and osteoprotegerin (OPG, also known as TNFRSF11B), which also functions as a soluble receptor [36,37]. TRAIL signaling is a promising target due to its ability to induce apoptosis independently of p53 status, making it particularly relevant for tumors resistant to DNA-damaging therapies [38].

However, tumor cells often evade apoptosis through the overexpression of decoy receptors, such as TNFRSF10C, TNFRSF10D, and OPG, which activate pro-survival pathways like NF-κB and contribute to metastasis and therapy resistance in many cancers, including in EOC [23,39]. Significant differences were found in OvarianTag™ genes related to TRAILR cell death. This offers a powerful approach by determining the expression profile of key genes in this pathway, enabling a more precise assessment of apoptotic signaling in EOC. By identifying tumors with a high expression of functional TRAIL receptors (TNFSF10A, TNFSF10B) versus those dominated by decoy receptors, OvarianTag™ could be a biomarker to predict sensitivity to TRAIL-based therapies. Furthermore, tumors with elevated decoy receptor expression may benefit from strategies targeting their downregulation, restoring apoptotic susceptibility, and improving therapeutic outcomes. With the growing understanding of apoptotic mechanisms in EOC, leveraging OvarianTag™ as a biomarker panel could enable a more personalized therapeutic approach, guiding the use of targeted therapies that enhance tumor cell death and overcome resistance mechanisms.

Furthermore, changes in genes involved in apoptosis signaling, such as TNFRSF10B, can lead to increased cancer cell death via alternative pathways like necroptosis [40,41]. Necroptosis is an emerging form of programmed cell death occurring via active and well-regulated necrosis, distinct from apoptosis morphologically and biochemically. Necroptosis is mainly unmasked when apoptosis is compromised in response to tumor necrosis factor-alpha. Unlike apoptotic cells, which are cleared by macrophages or neighboring cells, necrotic cells release danger signals, triggering inflammation and exacerbating tissue damage [42]. By validating the role of necroptosis pathways in treatment-resistant EOC, OvarianTag™ opens the door to developing the development of innovative therapies targeting necroptosis mediators such as RIPK1, RIPK3, and MLKL. Additionally, leveraging drug repurposing strategies to modulate these pathways provides an accelerated and cost-effective route to clinical translation [43,44,45].

Several genetic signature markers have been associated with the prognosis. These signatures provide insights into the disease process and a wide range of potential therapeutic pathways based on the described signaling networks within gene sets. Commercially available molecular profiling assays, such as Oncotype DX™ and MammaPrint™, have been extensively studied for their prognostic and predictive value, particularly in breast cancer. Oncotype DX™ assesses the expression of a specific panel of genes to estimate the potential benefit of adjuvant chemotherapy in breast cancer [46]. At the same time, MammaPrint™ is used to stratify patients based on the risk of breast cancer recurrence, aiding in the selection of the most appropriate therapeutic strategy [47]. Their integration into clinical practice allows for more accurate risk assessment, guiding therapeutic choices and ultimately optimizing patient outcomes [12]. Our study identifies a specific genomic signature that offers valuable insights into EOC and can potentially enhance risk assessment and therapeutic decision-making. Similar to other genomic signatures initially designed for breast cancer that have proven informative in different malignancies, OvarianTag™ was primarily developed for EOC and could be further validated to establish its utility in refining prognostic assessment and guiding targeted therapeutic strategies across a broader spectrum of cancer types.

Nowadays, in the ovarian cancer context, CA125 is still the most used prognosis biomarker in practice [48]. CA125 is a mucin-type glycoprotein (MUC16) that is elevated in 83% of patients with ovarian cancer. However, the CA125 blood test is not an effective screening test when used alone, given that CA125 levels are only increased in 50% of stage I ovarian cancers and can also be increased in benign disorders, such as uterine fibroids, ovarian cysts, and other conditions such as liver disease and infections [31]. In contrast, OvarianTag™ utilizes gene expression analysis in apoptosis and necroptosis pathways to deliver actionable insights into tumor biology, chemotherapy resistance, and recurrence risk, thereby addressing critical unmet needs in precision medicine. Other authors have considered this strategy to overcome CA 125 challenges, such as the combination of CA125 and HE4 levels to predict the risk of ovarian cancer in patients with suspected benign ovarian tumors, RMI (Risk of Malignancy Index), and ROMA (Risk of Ovarian Malignancy Algorithm) [49]. Taken together, the combination of OvarianTag™ with CA125 measurements offers a more reliable and precise prognostic tool to distinguish patients at risk of recurrence within 12 months and the potential benefit of platinum-based chemotherapy. Furthermore, this approach allows clinicians to stratify patients more effectively, facilitating the earlier adoption of alternative therapies such as PARP inhibitors or apoptosis-inducing treatments [50,51,52,53].

Despite the promising findings of our study, several limitations must be acknowledged. First, as a retrospective study, our analysis is inherently subject to selection bias and relies on pre-existing clinical and molecular data, which may limit the generalizability of our results. Additionally, while OvarianTag demonstrates strong potential as a predictive and prognostic tool, further prospective validation in independent cohorts is essential to confirm its clinical utility and reproducibility. Furthermore, while we propose that integrating OvarianTag™ with CA125 measurements enhances predictive accuracy, additional studies are required to establish standardized thresholds and evaluate its performance in real-world clinical settings.

## 4. Methods and Materials

### 4.1. Study Design and Participants

A retrospective analysis included patients who attended the Mario Penna Institute between 2008 and 2016. This study was conducted in two phases. First, 45 women were divided into the following three groups: EOC (n = 16), ovarian serous cystadenoma (n = 11), and normal ovary (NO) tissues (as control) (n = 18). Fresh-frozen tissue was collected during the surgery. In the second phase, 55 EOC patients were included. The cohort was stratified: 71 formalin-fixed, paraffin-embedded (FFPE) tissue samples were obtained and pooled into primary and metastatic EOC.

Women over 18 without ovarian malignant neoplasia evidence underwent bilateral oophorectomy for solicitation or indication in postmenopausal women or for some surgical intervention that made it necessary to withdraw the organ, for example, active bleeding. The withdrawal of part or the whole ovary was not accomplished only for purposes of the study. In the EOC patient group, the samples were collected from the primary tumor, and a sample of metastatic tumor was collected when extra-pelvic disease above 1 cm was observed. The tumor staging was performed according to the recommendations of the International Federation of Gynecology and Obstetrics (FIGO) [54]. These patients were recruited from partner hospitals, and their participation in this research does not imply any changes to their proposed treatment.

All of the clinical features consistently documented in medical records across all groups, like histopathological characteristics of the tumors, the presence of ascites, age at menarche, and menopausal status, were collected. The study was conducted under the Institutional Ethics Committee for Research Involving Human Beings guidelines and approved under protocol CAAE: 00622412.3.0000.5135. All participants provided written informed consent before sample collection. The consent form included authorization to use biological material and the associated clinical data exclusively for research purposes.

### 4.2. Total RNA Extraction and Quality Analysis

RNA was extracted from five sections (10 mm thick) of each paraffin tissue sample (FFPE). Paraffin was removed by xylene extraction followed by ethanol washing. According to the manufacturer’s recommendations, total RNA was isolated from deparaffinized tissue using RecoverAll™ Total Nucleic Acid Isolation Kit for FFPE (Invitrogen, Carlsbad, CA, USA, cat # AM1975). The concentration of RNA samples and the 260/280 absorbance ratio were measured using the Nanodrop™ 2000 spectrophotometer (ThermoFisher Scientific-Waltham, MA, USA), and only samples with adequate concentrations and a 260/280 ratio between 1.8 and 2.1 were included for cDNA synthesis and downstream analyses.

The selection of FFPE samples for gene expression assays was consistent with the choice of ideal tissue blocks for standard immunohistochemistry assays, all analyzed by the same pathologist to avoid biases. Essentially, the block contains the most significant amount of ovarian carcinoma that is morphologically consistent with the submitted diagnosis.

### 4.3. Gene Expression Analysis

cDNA was synthesized using SuperScript™ IV VILO™ Master Mix with ezDNase™ Enzyme (Invitrogen, cat. # 11766050). Quantitative PCR (qRT-PCR) was performed using TaqMan™ fluorochrome-labeled linear probe array technology (Applied Biosystems, Foster City, CA, USA) on a 7500 Fast Real-Time PCR System (Applied Biosystems, Foster City, CA, USA) to determine relative gene expression, starting with a set of genes related to apoptosis and the necroptosis-regulated cell death pathway. All assays span intron sequences and so do not detect genomic DNA. The qRT-PCR was prepared using TaqPath™ qPCR Master Mix, CG (Applied Biosystems, cat. #4367659).

Cases 70, 63, and 68 are representative samples from the study cohort, including both normal and tumor ovarian tissues. The Appendix A shows qRT-PCR-derived values (Ct-based) for six candidate endogenous control genes (18S, ALAS, GAPDH, SDHA, TBP, and TFRC). Expression stability (M) and normalization factors were calculated using the GeNorm algorithm—(https://www.ciidirsinaloa.com.mx/RefFinder-master/ (accessed on 12 October 2018). Genes with lower M values are considered more stable. Based on this analysis, TATA-box binding protein-TBP (TBP) (Assay number: Hs00427620_m1-ThermoFisher Scientific) was identified as the most stable gene and was used as the internal control for normalization in all of the qRT-PCR assays throughout the study. The fold change of gene expression was calculated with the 2−ΔΔCT method as described by Livak and Schmittgen [55]. In this method, the Ct values of target genes were first normalized to an endogenous reference gene, and then to a calibrator sample (e.g., normal ovarian tissue), resulting in the ΔΔCt values.

Additionally, we compared our findings with the existing literature, which reports a lack of association between these genes and clinical outcomes.

### 4.4. Data Analysis and Developing an Intelligent System for Clinical Outcomes Prediction in EOC Patients

Data obtained from the gene expression assay and clinical data of patients were stored and analyzed with appropriate statistical tools using the program SPSS 19.0 for Windows. A *p*-value < 0.05 was considered statistically significant. To visualize and graphically represent the expression values of the genes studied in the patients, heatmaps were constructed using the “gplots” package (https://www.rdocumentation.org/packages/gplots/versions/3.2.0, accessed on 16 February 2025), using the default parameters for clustering and the “heatmap.2” function in the R software (version 3.6.3) [56].

Supervised machine learning methods were used in clinical, pathological, and molecular EOC patient data mining. Attribute selection methods were applied in datasets, searching for biomarkers important to the prognostic evaluation of and outcome prediction for EOC patients. Classifiers were generated using the tree-based and black-box algorithms [57]. Attributes not contributing to the classification model were excluded from the datasets based on the composition results expressed. For these constructions and the understanding of class relationships, C4.5 implementation was used. The validation and generalization of the method were done using Leave-One-Out Cross-Validation (LOOCV).

The classifier algorithm’s performance was evaluated based on negative predictive value (NPV), positive predictive value (PPV), sensitivity, specificity, negative and positive likelihood ratios, and accuracy. The second dataset was used to retrain the algorithm and assess its performance.

The discriminative performance of the binary classification model was evaluated using Leave-One-Out Cross-Validation (LOOCV). For each iteration, the model was trained on all samples except one, which was used for testing. The predicted probability for the positive class was recorded, and overall performance was assessed using the Receiver Operating Characteristic (ROC) curve and the Area Under the Curve (AUC) metric.

With the best attributes for data classification and the classifiers, a computational solution in software was created and is available on the website www.oncotag.com.br Based on the selected attributes and classifiers, we developed a proprietary computational solution, which is currently hosted on our institutional website (www.oncotag.com.br). The first step in building the software was related to gathering requirements and specifying functionalities and constraints. Each identified requirement was analyzed to resolve ambiguities, contradictions, and hidden or incomplete requirements. They were documented at the end of the requirements analysis, and the software coding phase was initiated.

## 5. Conclusions

Our study presents OvarianTag™ as an innovative genomic tool that leverages apoptosis and necroptosis pathway signatures to improve risk stratification and therapeutic decision-making in EOC. By identifying key molecular determinants of platinum resistance and tumor recurrence, OvarianTag™ may contribute to more personalized treatment strategies. While these findings are encouraging, future prospective validation and functional studies will be essential to confirm the clinical utility of these findings, ultimately improving patient outcomes in EOC and potentially other malignancies.

## Figures and Tables

**Figure 1 ijms-26-08393-f001:**
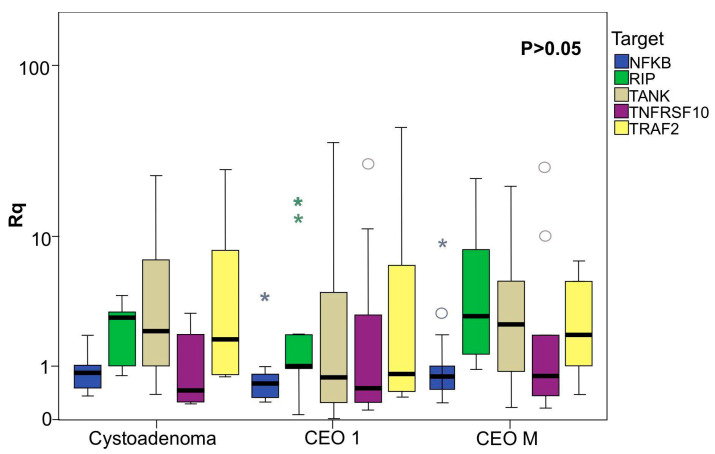
Association of TNFRSF10D/TRAIL-R4, NF-ΚB, RIPK1, and TRAF2 expression in ovarian tumors. The values represent the expression levels of each gene evaluated. The horizontal line indicates the median expression ratio, and the boxes demonstrate the interquartile range (25–75%). Percentile intervals from 10 to 90 are also shown. Differences between groups were assessed using the Kruskal–Wallis test. Asterisks indicate statistically significant differences between primary and metastatic EOC samples (*p* < 0.05).

**Figure 2 ijms-26-08393-f002:**
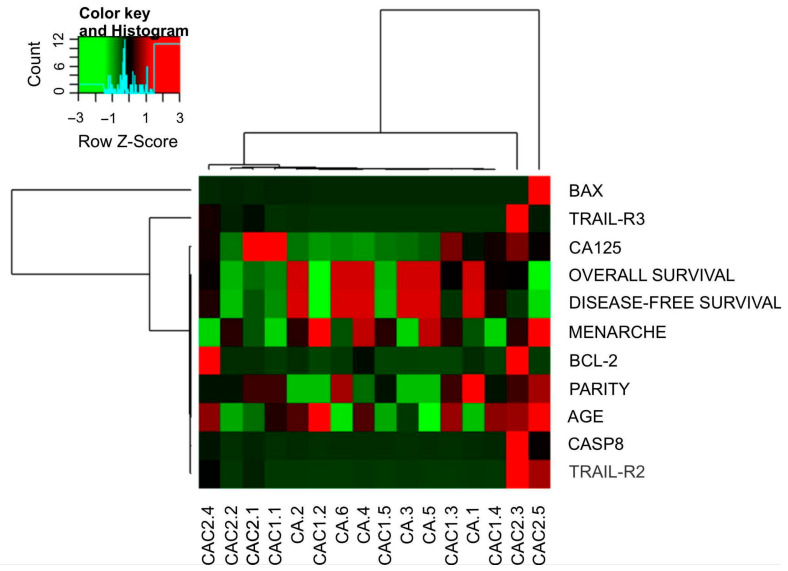
Heatmap showing the grouping of expression values of the genes studied in the groups of patients with primary EOC, metastatic EOC, and cystadenoma. It can be observed that the normalized expression values (Z-core) for each target by attribute corroborate the classifications presented by the decision tree method.

**Figure 3 ijms-26-08393-f003:**
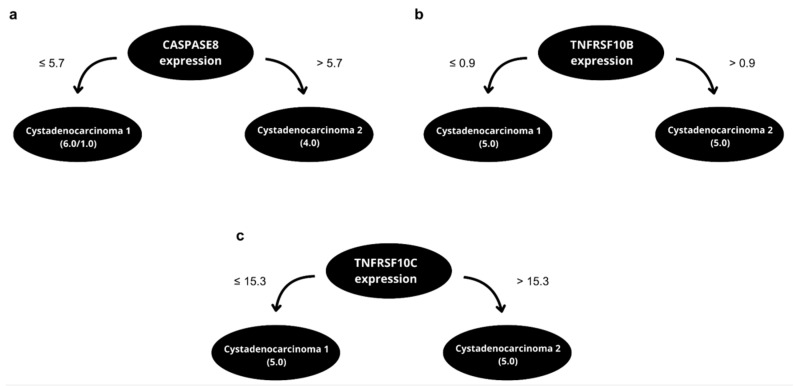
Decision trees constructed for the targets CASPASE8 (**a**), TNFRSF10B/TRAIL-R2 (**b**), and TNFRSF10C/TRAIL-R3 (**c**) using records from patient groups with early EOC, advanced EOC, and cystadenoma.

**Figure 4 ijms-26-08393-f004:**
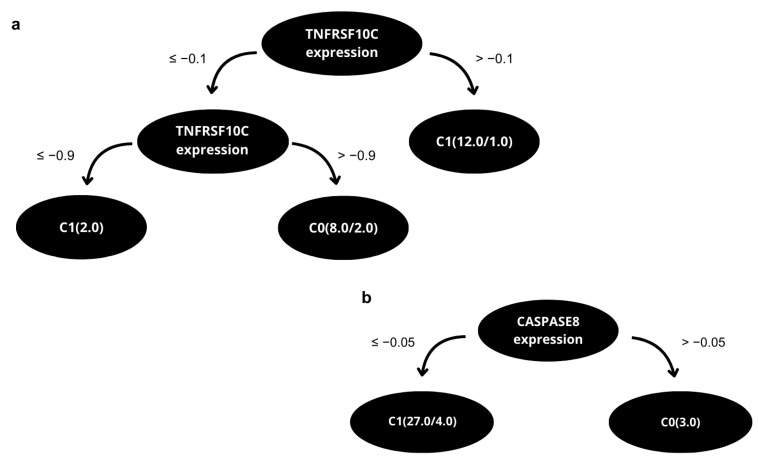
Decision trees for targets 2 (**a**) and 3 (**b**), constructed using records from patients with primary and metastatic DSC. These models were applied to predict clinical outcomes.

**Figure 5 ijms-26-08393-f005:**
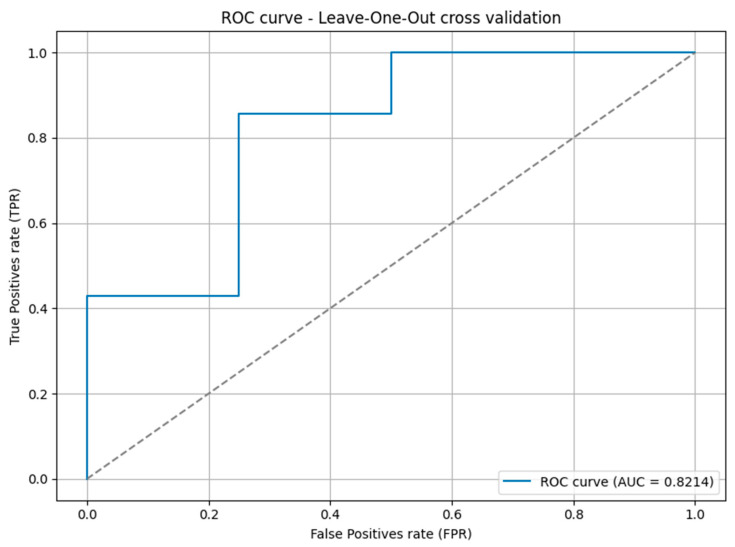
Receiver Operating Characteristic (ROC) curve obtained via Leave-One-Out Cross-Validation (LOOCV). The curve illustrates the trade-off between true positive rate (TPR) and false positive rate (FPR) across different classification thresholds. The calculated Area Under the Curve (AUC) was 0.8214, reflecting the classifier’s discriminative capacity.

**Figure 6 ijms-26-08393-f006:**
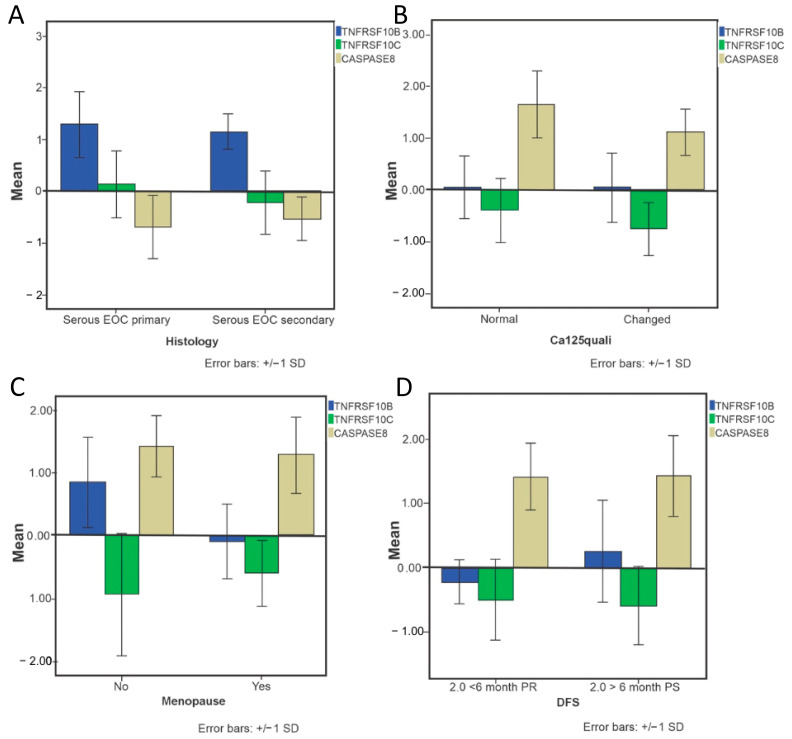
Association of OvarianTag™ panel target expression and clinical-pathological characteristics of patients. (**A**) histology, (**B**) normal (<35 U/mL) and altered (>35 U/mL) ca125 levels, (**C**) menopause, and (**D**) progression-free survival in patients with ovarian serous epithelial tumors. The values shown represent the expression levels of each target evaluated. Differences between groups were assessed by the Mann–Whitney U test.

**Table 1 ijms-26-08393-t001:** The clinical and pathological patient’s characteristics.

		1st Phase		2nd Phase	
	Normal ovary	Ovarian serous cystadenoma	Cystadenocarcinoma ovarian serous	Cystadenocarcinoma ovarian serous	*p* value
	n = 18	n = 11	n = 16	n = 55	
Age (Years)	50.61 ±2.13	54.3 ± 5.1	59.1 ± 2.7	58.7 ± 11.2	
Parity	2.83 ± 0.44	2.3 ± 0.8	1.9 ± 0.3	1.6 ± 1.3	0.128
**CA-125 (U/mL)**					
CA < 35 (U/mL)		11 (100%)	4 (25%)	12 (22.2%)	
CA > 35(U/mL)		0	12 (75%)	42 (77.8%)	
**Cytoreduction**					
Excellent (<1 cm)			7 (43.8%)	21 (46.7%)	
Suboptimal (>1 cm)			9 (56.33%)	24 (53.33%)	
**Degree of tumor** **differentiation**					
G2			6 (37.5%)	8 (22.2%)	
G3			10 (62.5%)	28 (77.8%)	
**Staging**					
I-II			5 (31.8%)	21 (41.2%)	
III-IV			11 (68.88%)	30 (58.88%)	
**Ascites**					
Yes			10 (62.5%)	30 (61.2%)	
No			6 (37.5%)	19 (38.8%)	
Menarche		12.8 ± 0.4	12.8 ± 3.4	13.05 + 1.36	0.991
**Menopause**					
Yes	18 (100%)	6 (54.5%)	13 (81.3%) *	38 (76%)	0.008
No	0	5 (45.5%)	3 (18.8%)	12 (24%)	

The values represent the mean ± standard error or n (percentage). The degree of tumor diffusion was classified as moderately differentiated (G2) and undifferentiated (G3). Monitoring was considered to be optimal when the residual tumor was <1 cm after surgery for tumor resection. The staging was performed according to the recommendations of FIGO (International Federation of Gynecology and Obstetrics), 2018 [20]. * *p* < 0.05 vs. normal ovary group.

**Table 2 ijms-26-08393-t002:** Results obtained and sensitivity and accuracy (LOOCV) of the decision tree classification method for the targets selected.

Analysis	Full Training	LOOCV	Rule
2017_10_16_Onco tag_histology_CA SP8_end	90%	80%	IF expression CASP8 <= 5.7 then early tumors else advanced tumors
2017_10_16_Onco tag_histology_TN RF10B_end	100%	90%	IF expression TNRFS10B <= 0.9 then cystadenoma 1 else cystadenoma 2
2017_10_16_Onco tag_histology_TN RF10C_end	100%	90%	IF expression TNRFSF10C <= 15.3 then cystadenoma 1 else cystadenoma 2

**Table 3 ijms-26-08393-t003:** Performance of the best classifiers in the analyzed datasets to determine the benefit of platinum-based treatment, the primary origin of the tumor, the risk of tumor recurrence, and the chance of progression-free survival for 6 months.

	The Benefit of Platinum-Based Treatment		Risk of Tumor Recurrence in 12 Months			
	Full Training	LOOCV	Full Training	LOOCV	Full Training	LOOCV
**Sensitivity**	100%	100%	88%	78%	100%	88%
**Specificity**	85%	82%	81%	80%	100%	71%
**Pos. Lik. Rat.**	67.5%	56%	46.7%	38.9%	-	31%
**Neg. Lik. Rat**	0	0	15%	28%	0	1.6%
**Pos. Pred. Va.**	42.9%	28.6%	70.0%	70.0%	100%	92%
**Neg. Pred.** **Va.**	100%	100%	92.9%	85.7%	100%	62.5%
**Accuracy**	86.7%	83.3%	83.3%	79.2%	100%	84.8%

LOOCV = Leave-One-Out Cross-Validation; Pos. Lik. Rat. = positive likelihood ratio; Neg. Lik. Rat. = negative likelihood ratio; Pos. Pred. Va. = positive predictive value; Neg. Pred. Va. = negative predictive value.

## Data Availability

The data presented in this study are available on request from the corresponding author. The data are not publicly available due to ethical restrictions related to patient confidentiality and the proprietary nature of the predictive algorithm and biomarker panel under industrial protection.

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
