# Peer review of "The OvarianTag™ Biomarker Panel Emerges as a Prognostic Tool to Guide Clinical Decisions in Cisplatin-Based Treatment of Epithelial Ovarian Cancer"

_ijms, 2025, doi:10.3390/ijms26178393_

Round 1
Reviewer 1 Report
Comments and Suggestions for Authors
1.The normal ovarian tissues were used in this study, and were there any ethical restrictions?
2.ROC curves and AUC values are critical metrics for evaluating machine learning performance, yet they were not mentioned in this study.
3.The descriptions of the x-axis and y-axis in Figures 1 and 2 are unclear. What do they specifically represent?
4.How were certain genes mentioned in lines 224-225 excluded during the construction of the machine learning models? The feature selection process lacks sufficient detail.
Author Response
For research article
|
Response to Reviewer 1 X Comments |
||
|---|---|---|
|
1. Summary |
|
|
Thank you very much for taking the time to review this manuscript. Please find the detailed responses below and the corresponding revisions/corrections highlighted/in track changes in the resubmitted files.
|
2. Questions for General Evaluation |
Reviewer’s Evaluation |
Response and Revisions |
|
Does the introduction provide sufficient background and include all relevant references? |
Can be improved |
Additional relevant references have been incorporated to strengthen the background. |
|
Are all the cited references relevant to the research? |
Must be improved |
Also, additional references have been incorporated to ensure alignment with the research focus and expanded the justification for marker selection and decision-making algorithms. |
|
Is the research design appropriate? |
Must be improved |
We have revised the section describing the study design to make the information clearer and more accessible to the reader. |
|
Are the methods adequately described? |
Must be improved |
We also included additional details regarding the selection criteria for the control group and addressed the ethical aspects of the study. |
|
Are the results clearly presented? |
Must be improved |
We have revised the manuscript to improve the clarity of the results presentation. Tables and figures were reviewed and adjusted where necessary, and the text was edited to ensure a clearer and more concise description of the findings. |
|
Are the conclusions supported by the results? |
Must be improved |
The conclusion has been revised to adopt a more balanced tone, acknowledging the study’s limitations and avoiding strong claims not yet supported by clinical validation.
|
- Point-by-point response to Comments and Suggestions for Authors
We also included additional details regarding the selection criteria for the control group and addressed the ethical aspects of the study.
Comments 1: The normal ovarian tissues were used in this study, and were there any ethical restrictions?
Response 1: Thank you for your question. There are many ethical restrictions to select normal ovarian group for our study. We discuss this point in the text "Women over 18 without ovarian malignant neoplasia evidence have undergone bilateral oophorectomy for solicitation or indication in postmenopausal women or for some surgical surgery intervention that made it necessary the withdrawal of the organ, for example, active bleeding. The withdrawal of part or the whole ovarian was not accomplished only for study purposes. In the EOC patients group, the samples were collected from the primary tumor, and a sample of metastatic tumor was collected when extra- pelvic disease above 1 cm was observed. The tumor staging was performed according to the recommendations of the International Federation of Gynecology and Obstetrics (FIGO)” in lines 113-121.
Comments 2: ROC curves and AUC values are critical metrics for evaluating machine learning performance, yet they were not mentioned in this study.
Response 2: Thank you for your observation regarding the ROC analysis. We agree that evaluating the model's discriminative performance using ROC and AUC metrics is essential. To address this, we applied Leave-One-Out Cross-Validation (LOOCV), recording the predicted probabilities for each held-out instance. We then computed the ROC curve and corresponding AUC using the aggregated results. This approach provides a more accurate and unbiased estimate of model performance on unseen data, particularly important for small datasets. The updated figure and methodological description have been included in the revised manuscript.
Comments 3: The descriptions of the x-axis and y-axis in Figures 1 and 2 are unclear. What do they specifically represent?
Response 3: Thank you for pointing this out. We agree with this comment and have made the necessary corrections. Specifically, we altered the axes and the resolutions of the figures.
Comments 4: How were certain genes mentioned in lines 224-225 excluded during the construction of the machine learning models? The feature selection process lacks sufficient detail.
Response 4 :We thank the reviewer for this insightful comment. The feature selection process was implicitly handled by the J48 algorithm itself, which implements the C4.5 decision tree method. During model construction, J48 recursively selects the most informative attributes based on information gain to build the tree, thus excluding features that do not contribute significantly to classification performance. In our study, although all the mentioned clinical and molecular attributes were initially provided as input, only three features were retained in the final tree structure. These features were selected by the algorithm based on their ability to optimize the classification of patient outcomes. To ensure transparency, all input features and their respective contributions to classification are listed in Supplementary Material 03. We have updated the manuscript to clarify this process accordingly.”

Reviewer 2 Report
Comments and Suggestions for Authors
The article titled "The OvarianTag™ Biomarker Panel Emerges as a Prognostic Tool to Guide Clinical Decisions in Cisplatin-Based Treatment of Epithelial Ovarian Cancer" presents the development and validation of the OvarianTag™ biomarker panel. This panel integrates apoptosis and necroptosis pathways to predict chemotherapy benefit and disease progression in epithelial ovarian cancer (EOC) patients. The study was analyzed retrospectively and involved two phases, using qRT-PCR and machine learning algorithms to identify relevant prognostic markers. The results showed significant differences in gene expression levels and high accuracy in predicting platinum response and recurrence risk. The authors conclude that OvarianTag™ has the potential to guide personalized treatment strategies and improve clinical decision-making.
The study offers a new and innovative approach to predicting chemotherapy response and disease progression in EOC. The findings in the manuscript clearly have significant clinical implications, potentially guiding personalized treatment strategies and improving patient outcomes in EOC. Overall, the manuscript is well-organized, with clear sections and a logical flow.
However, to enhance the manuscript, the authors should address a few points:
- The discussion does not address potentially confounding factors that may influence gene expression and clinical outcomes, such as patient comorbidities and treatment variations. Commentary on the above is required. Additionally, while the authors acknowledge certain limitations of the study, they should also acknowledge that this study lacks external validation in diverse populations, countries and clinical settings, which is and will be crucial for establishing the panel's broader applicability.
- The authors should add additional information to the methodology section as well as figure and table legends and should proofread the manuscript to ensure that all the legend for all data presented is comprehensive. Specific examples are listed below:
- Methodology: The authors indicate in line 102 that a retrospective analysis was performed. What does that encompass? Was all the experimental data collected prospectively, and only the analysis and machine learning/algorithm development performed retrospectively? Or was the RNA extraction, gene expression performed on bio banked samples?
- Table and Figure Legends: Additional information is required in supplementary material 1 legend indicating if data normalization was performed only on a subset of samples indicated by the case number or if the cases listed are representative. Are 70, 63 and 68 all tumor samples or are any of these normal samples? The legend needs to be re-written with a lot more details. Also, what do the numbers under each gene for each case represent? Are these calculated quantities?
- Supplementary material 2: The Y-axis markings are missing from the DFS graph.
- Results: The mean ca-125 level was 1927.4 U/mL +3856.4, DFP 15.34 195 +6.8, and OS 23.7 +37 of the patients included in the second phase of this study (lines 195, 196). Please explain why these results shouldn’t be represented as median instead of mean given the high standard deviation.
Line 140: select instead of selected
Line 142: repetition of ‘to select endogenous control’ since already mentioned in lines 139 and 140.
Line 173: software was available and ‘is’ available
Line 181: status of menopause in place of frequency of menopause
Line 186: menopause status in place of concerning menopause
Line 299: provide the full form of DD-containing receptors since not provided in the manuscript earlier
Line 300: I believe the alternative decoy receptor mentioned here should be TNFSF10C instead of TNFSF10B
Line 350: The should be small-caps
Author Response
For research article
Response to Reviewer 2 X Comments
1. Summary
Thank you very much for taking the time to review this manuscript. Please find the detailed responses
below and the corresponding revisions/corrections highlighted/in track changes in the resubmitted files.
2. Questions for General
Evaluation
Does the introduction provide
sufficient background and include
all relevant references?
Reviewer’s Evaluation Response and Revisions
Yes Thank you for your
positive feedback.
Is the research design appropriate? Yes Thank you for your
positive feedback.
Are the methods adequately
described?
Can be improved We have revised the section
describing the study design
to make the information
clearer and more accessible to
the reader.
Are the results clearly presented? Can be improved We also included additional
details regarding the
selection criteria for the
control group and addressed
the ethical aspects of the
study.
Are the conclusions supported by
the results?
Yes Thank you for your
positive feedback.3. Point-by-point response to Comments and Suggestions for Authors
The article titled "The OvarianTag™ Biomarker Panel Emerges as a Prognostic Tool to Guide Clinical
Decisions in Cisplatin-Based Treatment of Epithelial Ovarian Cancer
" presents the development and
validation of the OvarianTag™ biomarker panel. This panel integrates apoptosis and necroptosis
pathways to predict chemotherapy benefit and disease progression in epithelial ovarian cancer (EOC)
patients. The study was analyzed retrospectively and involved two phases, using qRT-PCR and machine
learning algorithms to identify relevant prognostic markers. The results showed significant differences in
gene expression levels and high accuracy in predicting platinum response and recurrence risk. The
authors conclude that OvarianTag™ has the potential to guide personalized treatment strategies and
improve clinical decision-making.
The study offers a new and innovative approach to predicting chemotherapy response and disease
progression in EOC. The findings in the manuscript clearly have significant clinical implications,
potentially guiding personalized treatment strategies and improving patient outcomes in EOC. Overall,
the manuscript is well-organized, with clear sections and a logical flow.
However, to enhance the manuscript, the authors should address a few points:
Comment 1: The discussion does not address potentially confounding factors that may influence gene
expression and clinical outcomes, such as patient comorbidities and treatment variations. Commentary on
the above is required.
Response 1: Thank you for your question. We agree with the points addressed. We reviewed the patients’
medical records in search of potential confounding factors; however, comorbidities were not consistently
documented across all groups, limiting our ability to include them uniformly in the analysis. Therefore,
we focused on variables more reliably reported and potentially relevant to the histopathological
characteristics of the tumors, presence of ascites, age at menarche, and menopausal status. Among these,
only menopausal status showed a statistically significant difference between groups, and this was
highlighted in the discussion (line 299-303): “Considering the potentially relevant factors that could
influence gene expression and clinical outcomes, we evaluated clinical features consistently documented
across all groups in medical records, and only menopausal status showed a statistically significantdifference between groups. This evidence permitted us to affirm that the OvarianTag™ addresses these
challenges by identifying gene expression patterns associated with platinum resistance, highlighting
TNFRSF10B, TNFRSF10C, and CASP8 as key attributes in its predictive algorithm.
”
Comments 2: Additionally, while the authors acknowledge certain limitations of the study, they should
also acknowledge that this study lacks external validation in diverse populations, countries and clinical
settings, which is and will be crucial for establishing the panel's broader applicability.
Response 2: Thank you for your feedback. We acknowledge that the study lacks external validation in
diverse populations, countries, and clinical settings, which is and will be crucial for establishing the
panel's broader applicability. These analyses are in progress, as a subsequent study by the Brazilian
National Research Council grants (APQ-406318/2024-7), however, we believe it is relevant that initial data
be published.
Comments 3: The authors should add additional information to the methodology section as well as
figure and table legends and should proofread the manuscript to ensure that all the legend for all data
presented is comprehensive. Specific examples are listed below:
Methodology: The authors indicate in line 102 that a retrospective analysis was performed. What does
that encompass? Was all the experimental data collected prospectively, and only the analysis and machine
learning/algorithm development performed retrospectively? Or was the RNA extraction, gene expression
performed on bio-banked samples?
Thank you for your feedback. In response, we have revised the section describing the study design to
make the information clearer and more accessible to the reader. We also included additional details
regarding the selection criteria for the control group and addressed the ethical aspects of the study. We
believe these revisions improve the overall transparency and clarity of the research design.
Comments 4: Table and Figure Legends: Additional information is required in supplementary material 1
legend indicating if data normalization was performed only on a subset of samples indicated by the case
number or if the cases listed are representative. Are 70, 63 and 68 all tumor samples or are any of these
normal samples? The legend needs to be re-written with a lot more details. Also, what do the numbers
under each gene for each case represent? Are these calculated quantities?Response 4: We agree that the legend of Supplementary Material 01 required additional clarification. The
listed cases (70, 63, and 68) are representative samples used to evaluate the stability of candidate reference
genes for normalization, including both tumor and normal ovarian tissues. The values under each gene
represent raw Ct-derived measurements from qRT-PCR assays. Using the GeNorm algorithm, we
calculated the M values (expression stability) and normalization factors for each candidate gene. TBP was
identified as the most stable reference gene and was used for normalization in all subsequent gene
expression analyses in the study. We have revised the legend of the Figure 1 (supplementary material) to
include these details for clarity: “Cases 70, 63, and 68 are representative samples from the study cohort,
including both normal and tumor ovarian tissues. The table shows qRT-PCR-derived values (Ct-based)
for six candidate endogenous control genes (18S, ALAS, GAPDH, SDHA, TBP, and TFRC). Expression
stability (M) and normalization factors were calculated using the GeNorm algorithm. Genes with lower M
values are considered more stable. Based on this analysis, TBP was identified as the most stable gene and
was used as the internal control for normalization in all qRT-PCR assays throughout the study.
”
Comments 5: Supplementary material 2: The Y-axis markings are missing from the DFS graph.
Response 5: Thank you for your observation. We acknowledge the omission of Y-axis markings in the
disease-free survival (DFS) graph presented in Supplementary Material 2. We will correct this by adding
appropriate Y-axis labels to represent the proportion of patients free of disease (e.g., 0.0 to 1.0), ensuring
the graph is fully interpretable. The revised supplementary material will be included in the updated
submission.
Comments 6: Results: The mean ca-125 level was 1927.4 U/mL +3856.4, DFP 15.34 195 +6.8, and OS 23.7
+37 of the patients included in the second phase of this study (lines 195, 196). Please explain why these
results shouldn’t be represented as median instead of mean given the high standard deviation.
Response 6: We acknowledge the high standard deviation observed in CA-125 levels, disease-free
progression (DFP), and overall survival (OS). However, these values reflect the natural biological
variability of human samples in a clinical cohort. Given the heterogeneous nature of epithelial ovarian
cancer and patient responses, we chose to represent the data using mean ± standard deviation, in
alignment with the assumptions of our statistical analyses and to reflect the overall distribution of the
cohort.
Comments on the Quality of English LanguagePoint 1: Line 140: select instead of selected
Response 1: Adjustments made as requested.
Point 2: Line 142: repetition of ‘to select endogenous control’ since already mentioned in lines 139 and
140.
Response 2: Adjustments made as requested.
Point 3: Line 173: software was available and ‘is’ available.
Response 3: Adjustments made as requested.
Point 4: Line 181: status of menopause in place of frequency of menopause
Response 4: Adjustments made as requested.
Point 5: Line 186: menopause status in place of concerning menopause.
Response 5: Adjustments made as requested.
Point 6: Line 299: provide the full form of DD-containing receptors since not provided in the manuscript
earlier.
Response 6: Adjustments made as requested.
Point 7: Line 300: I believe the alternative decoy receptor mentioned here should be TNFSF10C instead of
TNFSF10B
Response 7: Thank you for your careful observation. You are absolutely right — the correct alternative
decoy receptor is TNFSF10C, not TNFSF10B. We apologize for the oversight, and the correction was been
incorporated into the revised manuscript in the line .359.
Point 8: Line 350: The should be small-caps
Response 8: Adjustments made as requested.

Reviewer 3 Report
Comments and Suggestions for Authors
Reviewer Comments
- Please provide more details about participant consent in the ethics statement within the Methods section.
- The majority of identified mRNAs with altered expression cannot be validated using datasets. While this issue has been thoroughly addressed, concerns about data quality persist. The authors should provide additional details on data processing, specifically the methods used for normalizing raw data.
- The authors should address the quality control of their RNA isolation, particularly when extracting RNA from FFPE tissues. Since formalin fixation alters nucleic acids and complicates the isolation of high-quality RNA for genetic profiling, further details on quality control measures should be provided.
- Could the authors provide further details on their diagnoses of risk stratification in EOC, including any potential delays in diagnosis? Delayed diagnosis is often associated with higher mortality rates, and additional insight into this aspect would be valuable.
- I would recommend some minor revisions, including a reduction and more focused discussion, along with a slight expansion on apoptosis and necroptosis, a little bit more clinical information.
Author Response
For research article
Response to Reviewer 3 X Comments
1. Summary
Thank you very much for taking the time to review this manuscript. Please find the detailed responses
below and the corresponding revisions/corrections highlighted/in track changes in the resubmitted files.
2. Questions for General
Evaluation
Does the introduction provide
sufficient background and include
all relevant references?
Are all the cited references
relevant to the research?
Is the research design appropriate? Reviewer’s Evaluation Response and Revisions
Yes Thank you for your
positive feedback.
Are the methods adequately
described?
Are the results clearly presented? Are the conclusions supported by
the results?
Yes Thank you for your
positive feedback.
Yes Thank you for your
positive feedback.
Yes Thank you for your
positive feedback.
Yes Thank you for your
positive feedback.
Yes Thank you for your
positive feedback.
3. Point-by-point response to Comments and Suggestions for Authors
Comments 1: Please provide more details about participant consent in the ethics statement within the
Methods section.Response 1: Thank you for your suggestion. We have revised and expanded the ethics statement in the
“Methods section” to clarify that all participants provided written informed consent before sample
collection, including permission to use biological samples under the approved ethics protocol (CAAE:
00622412.3.0000.5135). We changed “The study was performed following the Institutions' Ethical
Committee for Research in Human Beings guidelines. Informed consent was obtained from all patients
involved (CAAE: 00622412.3.0000.5135)” to “The study was conducted under the Institutional Ethics
Committee for Research Involving Human Beings guidelines and approved under protocol CAAE:
00622412.3.0000.5135. All participants provided written informed consent before sample collection. The
consent form included authorization for using biological material and associated clinical data exclusively
for research purposes.
”
Comments 2: The majority of identified mRNAs with altered expression cannot be validated using
datasets. While this issue has been thoroughly addressed, concerns about data quality persist. The authors
should provide additional details on data processing, specifically the methods used for normalizing raw
data.
Response 2: Thank you for this valuable feedback. In the submitted manuscript, the initial dataset
included all listed genes and clinical variables. During the model training, the algorithm automatically
selected the best features that achieved the classification performance based on accuracy in both full
training and leave-one-out cross-validation (LOOCV). This process was automated. After this, guided by
performance metrics, we selected the best decision tree for outcome predictions. These details are already
presented in the submitted version: the complete results for all tested features, including those that were
excluded, are provided in Supplementary Material 03, and the final selected features are summarized in
Table 2. We revised the main text to emphasize this information more clearly. In line 273-277, we added
“ After evaluating classification performance, only three features were retained based on their superior
predictive value. . All tested features, including those excluded, and their respective performance metrics,
are detailed in Supplementary Material 03.
” In response to your comment regarding data processing and
normalization, we have added details on the normalization method used for mRNA quantification.
Specifically, gene expression levels were normalized using the comparative Ct method (ΔΔCt). First, each
target gene’s Ct value was normalized to an endogenous reference gene (housekeeping gene) to obtain the
ΔCt value. Then, this ΔCt was normalized to a calibrator sample (e.g., normal ovarian tissue), resulting inthe ΔΔCt. The final relative expression was calculated using the formula 2^
-ΔΔCt. This method accounts
for variability in RNA input and reverse transcription efficiency, ensuring more reliable comparisons
across samples. The description of this normalization procedure was added in the Methods section (lines
175–186) “In this method, Ct values of target genes were first normalized to an endogenous reference
gene, and then to a calibrator sample (e.g., normal ovarian tissue), resulting in the ΔΔCt values.
” to
improve transparency regarding data processing.
Comments 3: The authors should address the quality control of their RNA isolation, particularly when
extracting RNA from FFPE tissues. Since formalin fixation alters nucleic acids and complicates the
isolation of high-quality RNA for genetic profiling, further details on quality control measures should be
provided.
Response 3: Thank you for this important comment. We recognize the challenges associated with RNA
isolation from FFPE tissues due to formalin-induced crosslinking and fragmentation. To ensure RNA
quality, we measured both RNA concentration and purity using the Nanodrop™ 2000 spectrophotometer
(ThermoFisher Scientific). Only samples with sufficient quantity and acceptable purity (260/280
absorbance ratio between 1.8 and 2.1) were selected for cDNA synthesis and subsequent gene expression
analysis. This quality control step was essential to minimize bias from degraded or contaminated
samples. We have added this information to clarify our quality assurance procedures: “and only samples
with adequate concentration and a 260/280 ratio between 1.8 and 2.1 were included for cDNA synthesis
and downstream analyses.
” in the Materials and Methods section (lines 156-157).
Comments 4: Could the authors provide further details on their diagnoses of risk stratification in EOC,
including any potential delays in diagnosis? Delayed diagnosis is often associated with higher mortality
rates, and additional insight into this aspect would be valuable.
Response 4: Thank you for this important observation. Risk stratification in this study was based on
standard clinical variables (FIGO stage, tumor grade, ascites, CA-125 levels) in combination with gene
expression profiles. As this was a retrospective cohort, we did not have access to consistent data on the
time interval between symptom onset and diagnosis, and potential diagnostic delays were notsystematically recorded. Nonetheless, the data reflects real-world clinical practice and captures the
natural variability observed in routine patient management.
Comments 5: I would recommend some minor revisions, including a reduction and more focused
discussion, along with a slight expansion on apoptosis and necroptosis, a little bit more clinical
information.
Response 5: Thank you for your thoughtful suggestions. Minor revisions have been implemented as
recommended. We agree that a more concise and focused discussion improves clarity. However, we
maintained a more detailed description of the apoptosis and necroptosis pathways, as understanding the
biological roles of the involved markers is essential to contextualize their selection and performance as
potential biomarkers. This background supports the rationale for their association with prognosis and
therapeutic response in EOC, reinforcing their clinical relevance.
